# LncRNA and circRNA in Patients with Non-Alcoholic Fatty Liver Disease: A Systematic Review

**DOI:** 10.3390/biom13030560

**Published:** 2023-03-20

**Authors:** Qingmin Zeng, Chang-Hai Liu, Dongbo Wu, Wei Jiang, Nannan Zhang, Hong Tang

**Affiliations:** 1Center of Infectious Diseases, West China Hospital of Sichuan University, Chengdu 610041, China; 2Division of Infectious Diseases, State Key Laboratory of Biotherapy and Center of Infectious Disease, West China Hospital, Sichuan University, Chengdu 610041, China; 3National Center for Birth Defect Monitoring, Key Laboratory of Birth Defects and Related Diseases of Women and Children, Ministry of Education, West China Second University Hospital, Sichuan University, Chengdu 610041, China

**Keywords:** non-alcoholic fatty liver disease, long non-coding RNA, circular RNA, biomarkers

## Abstract

Non-alcoholic fatty liver disease (NAFLD) is currently the most common cause of chronic liver disease worldwide. Early identification and prompt treatment are critical to optimize patient management and improve long-term prognosis. Long non-coding RNA (lncRNA) and circular RNA (circRNA) are recently emerging non-coding RNAs, and are highly stable and easily detected in the circulation, representing a promising non-invasive approach for predicting NAFLD. A literature search of the Pubmed, Embase, Web of Science, and Cochrane Library databases was performed and 36 eligible studies were retrieved, including 18 on NAFLD, 13 on nonalcoholic steatohepatitis (NASH), and 11 on fibrosis and/or cirrhosis. Dynamic changes in lncRNA expression were associated with the occurrence and progression of NAFLD, among which lncRNA NEAT1, MEG3, and MALAT1 exhibited great potential as biomarkers for NAFLD. Moreover, mitochondria-located circRNA SCAR can drive metaflammation and its inhibition might be a promising therapeutic target for NASH. In this systematic review, we highlight the great potential of lncRNA/circRNA for early diagnosis and progression assessment of NAFLD. To further verify their clinical value, large-cohort studies incorporating lncRNA and circRNA expression both in liver tissue and blood should be conducted. Additionally, detailed studies on the functional mechanisms of NEAT1, MEG3, and MALAT1 will be essential for elucidating their roles in diagnosing and treating NAFLD, NASH, and fibrosis.

## 1. Introduction

Nonalcoholic fatty liver disease (NAFLD), characterized by the excess accumulation of fat in hepatocytes, is emerging as the most common cause of chronic liver disease in children and adults worldwide [1,2]. Its prevalence worldwide in the general population is about 30% and this is expected to continue to increase over the next 20 years, placing a heavy burden on socioeconomic and healthcare systems [3]. However, the responses of healthcare professionals and public health institutions to NAFLD remain weak and fragmented; therefore, it is crucial to popularize the concept and strengthen public awareness and concern about NAFLD [4,5]. In addition, NAFLD is a multisystem disease and is often accompanied by obesity, type 2 diabetes, and other metabolic syndromes, and a key challenge is to identify those at the highest risk for NAFLD in large populations [6]. Nonalcoholic steatohepatitis (NASH) is the progressive form of NAFLD and it can progress to fibrosis, cirrhosis, and hepatocellular carcinoma (HCC). NASH phenotypes include macrovesicular steatosis, hepatocyte ballooning, and lobular inflammation, with or without peri-sinusoidal fibrosis [7]. Given that NASH-related cirrhosis is predicted to become the leading indication for liver transplantation and that fibrosis is the major determinant of clinical outcomes, identifying patients at higher risk of progressive liver disease, including NASH and advanced fibrosis, is critical in managing NAFLD [8,9,10]. Early access to reliable, non-invasive diagnostic tools is needed to identify patients at different stages of disease. No practicable drug is currently recommended for the treatment of NALFD, and the tailoring of therapeutic strategies to patients’ disease drivers is an issue that urgently needs to be addressed.

In recent years, with the development of high-throughput sequencing technologies, non-coding RNAs (ncRNAs), including long non-coding RNAs (lncRNAs), circular (circRNAs), and microRNAs (miRNAs), have emerged as important regulators in the pathogenesis of NAFLD. Previous reviews have extensively discussed miRNAs’ expression profiles and regulatory functions in NAFLD; therefore, these topics are not discussed here [11,12]. LncRNAs are transcripts with more than 200 nucleotides, which exert functions in transcriptional and post-transcriptional regulation, epigenetic modifications, and disease development [13,14]. CircRNAs are covalently closed single-stranded loop biomolecules without a 5′ cap or a 3′ poly (A) tail. CircRNAs regulate the gene expression at transcription and post-transcription levels by acting as miRNAs and protein sponges, as well as protein templates [15]. Previous studies have shown that these differentially expressed ncRNAs are implicated in the etiology of NAFLD and could possibly be the key mediators in its pathogenesis, involving the regulation of hepatic gluconeogenesis and lipogenesis, insulin resistance, oxidative stress, metabolic inflammation, regeneration, and fibrogenesis [16,17,18]. These molecules, as potential biomarkers for NAFLD diagnosis and staging, are gradually gaining the attention of researchers. In addition, various approaches, including RNA interference and overexpression techniques, have been developed to target these molecules, showing certain strengths and limitations [19]. However, these data have been obtained mainly from cell and animal models, and extensive validation in human patients is essential. In this review, all the currently published data on human NAFLD were included. We aimed to identify and pick out potential biomarkers by analyzing the differences in the expression of lncRNA and circRNA in the NAFLD disease spectrum, as well as analyzing their diagnostic accuracy in differentiating healthy people and patients with nonalcoholic fatty liver (NAFL), NASH, or fibrosis.

## 2. Methods

### 2.1. Literature Search Protocol and Search Strategy

We conducted a comprehensive search in Pubmed, Embase, Web of Science, and the Cochrane Library. The keywords and MeSH terms in our search strategy were as follows: lncRNA, long non-coding RNA, long ncRNA, circular RNA, circRNA, ciRNA, NAFLD, NASH, fatty liver, liver steatosis, and a combination of these MeSH terms. We also screened all articles referenced in these selected studies to identify additional articles.

### 2.2. Study Selection and Eligibility Criteria

The inclusion criteria were as follows: (a) studies that obtained lncRNA and/or circRNA expression profiles in patients with NAFLD or NASH; (b) studies that used liver tissue, serum, plasma, and blood as the samples; and (c) studies that used quantitative real-time PCR to measure lncRNA and/or circRNA expression. The exclusion criteria were as follows: (i) duplicate reports; (ii) studies conducted in animals or cell lines; (iii) systematic reviews or meta-analyses; and (iv) case reports, comments, letters, and editorials. Conference abstracts without grouping information and sample sizes were excluded, as were abstracts representing full-text articles already included in this study.

### 2.3. Data Extraction

Two authors (QMZ and CHL) independently assessed the full-text articles for eligibility and collected key information in a standardized form. Extracted lncRNA and circRNA data were derived from the GEO database or the microarray and high-throughput sequencing (RNA-seq) dataset. Quantitative real-time PCR was used to validate the circulating or liver lncRNA and circRNA expression. Other key information included the study design and population, sample type (liver tissue, serum, and plasma) and size, lncRNA/circRNA expression direction (up- or downregulated), and fold-change. Discrepancies related to the included data were resolved through discussions with a third review author (WJ).

## 3. Results

### 3.1. Summary of Studies on the lncRNA and circRNA Expression Profile in NAFLD

A flowchart of the selection process of studies included in this systematic review is shown in Figure 1. We identified 685 studies, of which 38 met the eligibility criteria (including 44 lncRNAs and 11 circRNAs) in the systematic review: 18 on NAFLD, 13 on NASH, 11 on fibrosis and/or cirrhosis, 6 on NASH-related phenotypes (NASH grade, lobular inflammation, steatosis, and NAS score), and 2 on NAFLD-related hepatocellular carcinoma (NAFLD-related HCC) (Figure 2). Seven studies reported on diagnostic accuracy experiments. Several conference abstracts related to full-text articles already included in this review were not discussed further in this article.

### 3.2. Differentially Expressed lncRNA in Patients with NAFLD

Sixteen studies (including 22 lncRNAs and approximately 550 individuals) revealed differentially expressed lncRNAs in patients with NAFLD, with 15 lncRNAs being upregulated, with a mean fold-change of 3.26 (range: 1.32–8.30), and 6 lncRNAs being downregulated, with a mean fold-change of 0.43 (range: 0.13–0.74) (Table 1). ncRNA-dependent epigenetic reprogramming affects a variety of metabolic pathways and cellular processes in the liver, including hepatic glucose and lipid metabolism, oxidative stress, the inflammatory immune response, and even tumorigenesis. These ncRNA-mediated metabolic abnormalities ultimately contribute to NAFLD development and progression (Figure 3). Most lncRNAs accelerated or attenuated NAFLD progression mainly through sponging microRNA (e.g., uc.372, uc.333, CCAT1, lnc-SPARCL1-1:2, NEAT, PVT1, and MALAT1) [20,21,22,23,24,25] or by directly targeting related proteins. Most studies focused on the differentially expressed lncRNAs in the liver tissue with small sample sizes; five other studies investigated the lncRNA expression profile in serum/plasma; one study focused on lncRNA expressions in peripheral blood mononuclear cells (PBMCs). However, those studies exhibited a narrow focus on one aspect, and the corresponding expression and correlation analysis between liver tissue and serum/plasma is also indispensable. Zhang et al. identified that lncARSR both in serum and liver was significantly increased in NAFLD patients compared with healthy controls, which may be a novel candidate biomarker of NAFLD [26]. Two studies observed increased lncRNA NEAT1 expression in patients with NAFLD in serum and PBMCs, respectively. Furthermore, the AUC of NEAT1 in PBMCs for NAFLD diagnosis was 0.822, with a sensitivity of 86.47% and a specificity of 82.03%. However, information on its expression in different NAFLD stages and its possible mechanisms is scant. Further studies taking these issues into account are needed in order to provide new insights into its utility for the diagnosis of NAFLD.

Meanwhile, there is an urgent need to establish new biomarkers for NAFLD subtype identification and classification. Two studies revealed that the serum levels of lncPRYP4-3 and RP11-128N14.5 in patients with NAFLD were higher than those in healthy controls, and their high expression pattern showed significant consistency in both subtypes, including NAFL and NASH, suggesting that they may be used to distinguish NAFLD patients from healthy individuals [42,43]. In addition, lncPRYP4-4 expression is merely increased in NAFL, rather than NASH, although the question of whether it can be used as a diagnostic marker of early NAFLD requires further study [42]. Zhang et al. reported that increased serum PVT1 has good predictive value for distinguishing NAFLD patients from healthy individuals, with an AUC of 0.895, a sensitivity of 84.0%, and a specificity of 84.6% [25]. Another lncRNA, HCG18, was identified as a promising candidate biomarker in differentiating NAFLD patients from healthy individuals (AUC = 0.934, sensitivity: 82.8%, and specificity: 89.1%) [35]. Furthermore, three other studies collectively focused on lncRNA MEG3 expression in NAFLD. One study found that MEG3 expression was elevated 1.6- and 1.9-fold, respectively, in NAFLD and NASH livers [28]. The authors of another study also observed that MEG3 was markedly increased in NASH cirrhosis specimens [39]. In contrast, Zou et al. showed the opposite finding that MEG3 was downregulated in NAFLD and its reduction paralleled the severity of NAFLD [40]. A possible explanation for this inconsistency might be that the reactivation and induction of MEG3 in NAFLD and NASH patients is likely a compensatory mechanism. The hepatic endothelial senescence promotes obesity-induced insulin resistance, which is tightly regulated by the expression of MEG3 [28]. MEG3 has been found to maintain glucose homeostasis and insulin signaling by protecting the hepatic endothelium against cellular senescence in obesity [28]. Future studies should carefully examine the status and functions of MEG3 in NAFLD at different stages. Furthermore, expanding the sample size or increasing the detection of circulating levels may provide more evidence.

### 3.3. Differentially Expressed lncRNAs in Patients with NASH

NASH is the progressive form of NAFLD, with hepatic necroinflammation and the faster progression of fibrosis. Early identification of NASH, combined with appropriate interventions, will prevent disease progression. Seven studies identified differentially expressed lncRNAs in serum or liver samples from patients with NASH (Table 1). Park et al. found that plasma LeXis was independently associated with NASH, with acceptable diagnostic performance (AUC = 0.743, sensitivity: 54.3%, and specificity: 100%) [45]. Compared with healthy controls, lncSPARCL1-1:2 was upregulated in NAFLD, simple steatosis, and NASH, and lnc-SPARCL1-1:2 was an independent predictor of NASH. It was used to distinguish NASH patients from healthy controls with an AUC of 0.870, to identify NASH with simple steatosis with an AUC of 0.790, and to identify NASH cases with NAFLD cases with an AUC of 0.974 [23]. However, the number of patients included in these studies was relatively small. Moreover, no external validation was performed on the diagnostic performance of these lncRNAs for NASH. Future studies on this issue are therefore recommended.

The major features of NASH include steatosis, lobular inflammation, hepatocyte apoptosis, and ballooning. A growing body of evidence suggests that lncRNAs play an important role in these characteristics (Table 2). A study by Di Mauro confirmed that serum RP11-128N14.5 and TGFB2/TGFB2-OT1 were upregulated in patients with NAS ≥ 5 compared with those with NAS ≤ 4, and the liver RP11-128N14.5 and TGFB2/TGFB2-OT1 were upregulated in severe NAFLD patients (NAS score ≥ 5, F = 3) compared with mild NAFLD patients (NAS ≤ 4, F = 0) and controls [43]. Furthermore, Atanasovska et al. showed that lnc18q22.2 was positively correlated with NASH grade, NAS score, and lobular inflammation [46]. Additionally, Leti et al. identified three upregulated lncRNAs in liver samples of NAFLD patients with lobular inflammation and advanced fibrosis, with the strongest evidence in MALAT1 [47]. Moreover, greater liver MALAT1 abundance was observed in NAFLD patients with higher scores of ballooning degeneration, lobular inflammation, and the presence of fibrosis in Sookoian’s study, which was validated both in the discovery and replication set [41]. Another study revealed that hepatic LeXis, but not plasma LeXis, was negatively correlated with the degree of steatosis [45]. RP1191K9.1 (lncTNF), another lncRNA analyzed in Atanasovska’s study, was also positively associated with lobular inflammation in human liver samples [48]. Taken together, dynamic changes of lncRNAs are associated with NASH phenotype and progression, which may provide new insights for the development of targeted drugs for NASH.

### 3.4. Differentially Expressed lncRNAs in Patients with Advanced Fibrosis or Cirrhosis

The fibrosis stage is a major determinant of all-cause and liver-related mortality and the long-term prognosis of NAFLD. Assessing the severity of fibrosis and identifying patients at higher risk of advanced fibrosis is crucial. There has been significant interest in developing ncRNA-based (microRNA, lncRNA, and circRNA) biomarkers to identify advanced fibrosis in patients with NAFLD. LncRNAs have exhibited corresponding expression patterns in different fibrosis stages. Eleven studies identified differentially expressed lncRNAs in serum or liver samples from patients with advanced fibrosis or cirrhosis (Table 3). One study by Han et al. showed that both liver and plasma lncRNA GAS5 expression was positively correlated with the progression of liver fibrosis in 51 patients with NAFLD, and this was more apparent in plasma. As fibrosis progressed towards cirrhosis, however, plasma GAS5 was downregulated [49]. In addition, downregulated liver GAS5 in patients with liver cirrhosis was also confirmed in Yu’s study [50]. Another four studies revealed that the serum TGFB2/TGFB2-OT1, lnc-SPARCL1-1:2, lncRNA RABGAP1L-DT-206, MALAT1, Neat1, and HULC expression were significantly higher in patients with advanced fibrosis/cirrhosis (F = 3–4) than in those without it (F = 0–2) [23,34,43,47]. Moreover, the differential expression of TGFB2/TGFB2 OT1 in advanced fibrosis/cirrhosis and absent or mild-to-moderate fibrosis patients was also confirmed in an external validation cohort of 50 NAFLD patients, and it yielded a high predictive capability for subjects with advanced fibrosis (AUC = 0.797 and 0.786 in internal and external cohorts, respectively), with sensitivity values of 65% and 62.5%, and specificity values of 81.3% and 94.4% in internal and external cohorts, respectively [43]. Notably, the diagnostic accuracy can be improved by combining TGFB2/TGFB2-OT1 and FIB-4 or TGFB2/TGFB2-OT1 and LSM (AUC = 0.891 and 0.892, respectively). In addition, liver MALAT1, in Sookoian’s study, was associated with fibrosis, as evidenced by a greater abundance of MALAT1 in patients with fibrosis (F1–F4) than in those without it (F0) [41]. Gerhard et al. also identified three dysregulated liver lncRNAs (LINC01638, LINC01605, XLOC_003146, and RP11_20J153) in 10 patients with advanced fibrosis [51]. These results further support the idea that functionally relevant differences in lncRNA expression may contribute to the development of fibrosis in NAFLD, and those lncRNAs may potentially be exploited as novel strategies to discriminate NAFLD/NASH cases from advanced fibrosis; moreover, the combination of other variables and non-invasive scoring systems may increase the diagnostic accuracy. Future investigations, including the validation of independent cohorts and the exploration of associated mechanisms, will be essential to extend these findings.

### 3.5. Differentially Expressed lncRNAs in Patients with NAFLD-Related HCC

HCC is one of the most common malignancies worldwide [53]. Increasing evidence has demonstrated that the rate of NAFLD-related HCC is increasing in parallel with the obesity epidemic, and NAFLD is projected to become a major cause of HCC incidence and mortality [54]. Identifying patients with NAFLD who have a high HCC risk is one of the crucial clinical challenges in NAFLD management [55]. Recently, various HCC-related lncRNAs and circRNAs have been revealed to exhibit aberrant expression and participate in HCC tumorigenesis and progression. Two studies identified that differentially expressed lncRNAs may participate in the progression of NAFLD to HCC. The lncRNA FTX was downregulated and the lncRNA SNHG20 was upregulated in NAFLD-related HCC tumor tissues, and FTX supplement and SNHG20 silencing inhibited the conversion of NAFLD to HCC, which prompted out interest in explorng its potential as a future HCC risk predictor [56,57]. In addition, most HCC-related lncRNAs are present in body fluids, which are easy to detect and analyze, giving them the potential to be attractive biomarkers in liquid biopsy of HCC [58,59]. Thus, an in-depth understanding of lncRNA functions and its roles in the pathogenesis of HCC will provide new insights and novel tools for the early diagnosis and treatment of HCC.

### 3.6. Differentially Expressed circRNAs in Patients with NAFLD and NASH

Recent years have seen the circRNA field explode and diversify; however, research on NAFLD is still a relatively young field. Three studies, including two circRNAs in NAFLD and nine circRNAs in NASH, have identified differentially expressed circRNAs in patients with NAFLD and NASH. Cytoplasm-localized circRNA_0046367 and circRNA_0001805, as well as mitochondrial-localized circRNA SCAR and eight other mitochondrial circRNAs, were significantly downregulated in patients with NAFLD and/or NASH [37,38,44]. Furthermore, circRNA SCAR exhibited a continuous decline from simple steatosis to NASH cirrhosis, which could be attributed to its driving effect on metabolic inflammation [44]. The differing sublocalization of these circRNAs corresponds with their roles in the pathogenesis of NAFLD, suggesting that the expression and localization of these functional circRNAs may be associated with the occurrence and progression of NAFLD. The concept of circRNAs as biomarkers for NAFLD and NASH is conceptually appealing, but further studies are needed to clarify their functional mechanism in NAFLD and their differences in different stages of NAFLD, such as NAFL, NASH, NASH-related fibrosis/cirrhosis, and NASH-related HCC.

## 4. Discussion

LncRNA and circRNA exhibit cell-type- and tissue-specific expression patterns, and can be released into circulating blood, urine, and other body fluids, where they show high stability. Therefore, the application of these molecules as novel biomarkers of NAFLD shows great advantages and promise, and may eventually have a significant impact on clinical practice. We reviewed differentially expressed lncRNAs and circRNAs in liver tissue or serum of NAFLD patients from all published studies and summarized the changes in these ncRNAs’ expression profiles under different severities of NAFLD and evaluated the diagnostic performance of some lncRNAs for NAFLD. LncRNA NEAT1, MEG3, MALT1, and GAS5 have attracted the attention of many researchers, showing great potential as biological markers of NAFLD. Verifying the expression direction of these lncRNAs in our NAFLD cohort and our collaborator’s NAFLD cohort will be a focus of our future studies.

In the present review, NEAT1 expression levels were found to be upregulated in serum and PBMCs of NAFLD patients, and it was also upregulated in both NAFLD-related inflammation and advanced fibrosis, which was consistent with previous studies conducted in vitro and in vivo. Recent preclinical studies have revealed that NEAT1 is abundant in the liver and is a risk factor affecting NAFLD, which could promote the NAFLD progress by facilitating hepatic lipid accumulation [60,61,62]. We first summarized NEAT1 expression levels in NAFLD patients, and they showed good diagnostic performance. However, one of the limitations of these studies is their small sample sizes, and validation in larger cohorts is essential. In addition, the question of whether NEAT1 is associated with the severity and prognosis of NAFLD remains unclear. Therefore, further studies are needed to examine circulating and liver NEAT1 levels at different stages of NAFLD in large cohorts and explore the underlying mechanism of NEAT1 in NAFLD.

Interestingly, the expression of MEG3 varied among different study cohorts. Zou et al. observed downregulated MEG3 expression in NAFLD patients, which was in line with previous preclinical studies showing that MEG3 was downregulated in in vitro and in vivo models of NAFLD and was negatively related to lipogenesis-related genes, and that boosting MEG3 expression alleviated HFD-induced NAFLD progression [40,63]. Nevertheless, another two studies showed the opposite results. There are two likely causes for this discrepancy. First, the induction of MEG3 in NAFLD/NASH patients is likely a compensatory mechanism. NAFLD patients frequently present insulin resistance, which promotes the reactivation of MEG3 to maintain glucose homeostasis. Furthermore, MEG3 expression is also regulated by its target gene in a feedback-regulatory fashion. Second, the same lncRNA may play different roles in different liver diseases due to changes in the physiological environment. MEG3 maintains systemic glucose homeostasis independently of its effects on systemic inflammation, and it can also drive fibrosis and even cirrhosis via the promotion of bile acid synthesis and cholestasis [28,39]. Thus, the exact role of MEG3 in NAFLD pathogenesis and its possible multidimensional effects need to be further elucidated. Future studies must focus on the combined effect of lncRNA on multiple targets in the liver. Furthermore, further large-cohort studies on the MEG3 expression profile in circulation and its relationship with NAFLD disease stage and severity need to be undertaken.

MALAT1 expression levels were upregulated 1.73-fold in NAFLD versus healthy controls, 1.75-fold in NASH versus NAFL, 3.01-fold in ballooning degeneration, and 5-fold in fibrosis, which suggests that MALAT1 plays a significant role in triggering NAFLD and perpetuating the NASH phenotype. Previous studies have suggested that MALAT1 can regulate PPARα/CD36-mediated hepatic lipogenesis and lipid accumulation and promote tissue inflammatory immune response by activating the NF-κB signaling pathway [29,64]. It can therefore be assumed that MALAT1 is positively associated with the full spectrum of histologic severity. Dysregulated MALAT1 has only been confirmed in intrahepatic samples; however, evidence in circulation is still lacking. Thus, further studies will be required to reveal the circulating levels of MALAT1 and to explore its role in liver–serum communication. Furthermore, the cellular sources of MALAT1 and its expression levels in other chronic liver diseases remain to be examined. Plasma GAS5 expression increased with fibrosis progression but decreased in the case of cirrhosis. A compensation mechanism of GAS5 may be involved. Namely, GAS5 may increase to resist the progression of fibrosis. Previous studies have revealed that GAS5 inhabited liver fibrogenesis through competitively binding certain miRNAs [50,65]. Further studies on GAS5 expression in patients with fibrosis from F0-F4 and even patients with NASH could provide more definitive evidence. Taken together, MALAT1 and GAS5 were positively correlated with NASH grade and/or fibrosis, and thus may be candidate biomarkers for severe NAFLD (such as NASH, fibrosis) and may open up new frontiers for NAFLD drug development.

Remarkably, some ncRNAs were expressed inconsistently or even inversely in different tissue compartments, for example, miRNA-122 was dysregulated in intrahepatic samples but upregulated in serum, suggesting that ncRNAs may likely play other roles that vary according to the biological or disease-related context. In addition, this intracellular-extracellular expression paradox was also reflected during NAFLD progression [66]. Thus, periodic dynamic measurements of tissue and circulating ncRNA expression to track disease changes in real-time are greatly need. Notably, most previous studies have merely analyzed the functional activity and diagnostic performance of a single ncRNA, frequently failing to accurately identify and distinguish between NAFLD, NASH, and fibrosis. Exploring the co-regulatory networks and combined effects of these ncRNAs is crucial and recommended. Moreover, some ncRNAs may play different or even opposite roles in different liver diseases, and cell- and disease-specific ncRNAs seem to have greater advantages in finding applicable biomarkers for NAFLD. Mitochondria-located circRNA SCAR promotes steatosis-to-NASH progression through driving metaflammation and may serve as a promising biomarker and therapeutic target for NASH [44]. Therefore, future studies should focus on cell- and disease-specific ncRNAs and on the combined effect of various ncRNAs on multiple targets, which will help us develop novel and valuable biomarkers for NAFLD diagnosis.

LncRNA- and circRNA-based gain-of-function and loss-of-function approaches exert a profound impact on NAFLD pathophysiology and phenotype, suggesting the therapeutic potential of these ncRNA-based therapies. Most lncRNAs and circRNAs exhibit tissue-specific expression patterns and lower expression levels compared with mRNAs, which can be administered at lower doses with fewer undesired toxic effects, but the physiological effects reflected by low abundance are still controversial, suggesting that identifying the main functional ncRNA domains that specifically interact with other RNAs, DNA, or proteins is crucial. Currently developed targeting approaches, including expression vectors, delivery vehicles, small interfering RNAs (siRNAs), antisense oligonucleotides (ASO), and CRISPR-Cas9/13 systems, show great promise, but their safety and undesired off-target effects need to be improved and solved before clinical application [19,67]. Furthermore, lncRNAs are longer than circRNAs, which inevitably activates the immune response and increases delivery challenges. Moreover, the poor sequence conservation of lncRNAs may increase the complexity of their clinical translation to human disease. However, although the sequences of many lncRNAs differ between species, their structures are often conserved [67,68]. Therefore, an in-depth understanding of the functional structure of lncRNAs is necessary for the rational design of lncRNA-targeted therapy.

## 5. Conclusions

The current accumulated evidence indicates that functionally relevant lncRNAs and circRNAs play important regulatory roles in NAFLD biological pathways, and they hold promise as potential biomarkers for NAFLD disease risk assessment, diagnosis, treatment, and prognosis. However, the available data are mostly limited to preclinical models and need to be validated in a broader external cohort. In addition, further investigations on the challenge of low expression abundance in regard to detection methods, the difficulty in the identification of complex functional domains, and the multiplicity of liver pathophysiology are necessary in order to screen and identify available and accurate biomarkers for NAFLD. Given the cell- and tissue-specific expression patterns of circRNAs and lncRNAs, the development of circRNA and lncRNA profiling at the single-cell level has great potential, and spatial transcriptomics may provide another way to study RNA expression in complex tissues.

## Figures and Tables

**Figure 1 biomolecules-13-00560-f001:**
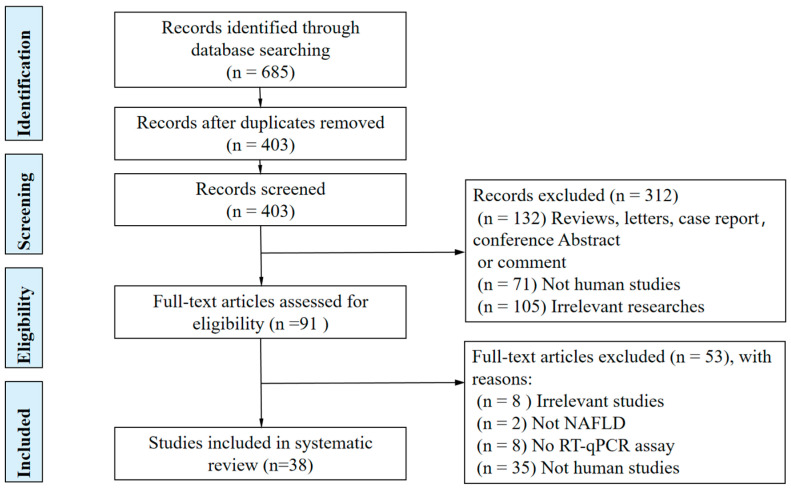
PRISMA flowchart of the study selection process.

**Figure 2 biomolecules-13-00560-f002:**
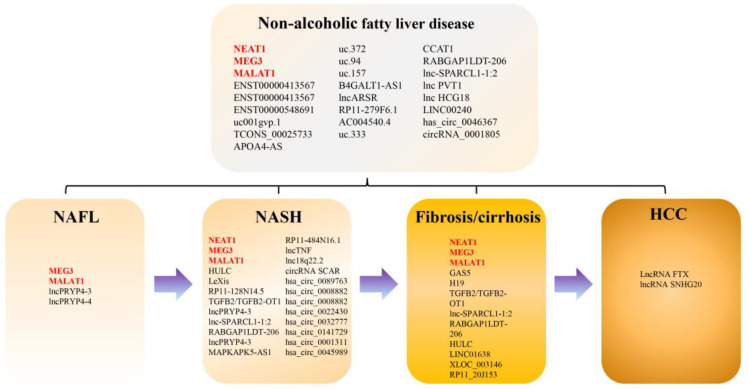
Deregulated lnRNAs and circRNAs in patients with NAFLD.

**Figure 3 biomolecules-13-00560-f003:**
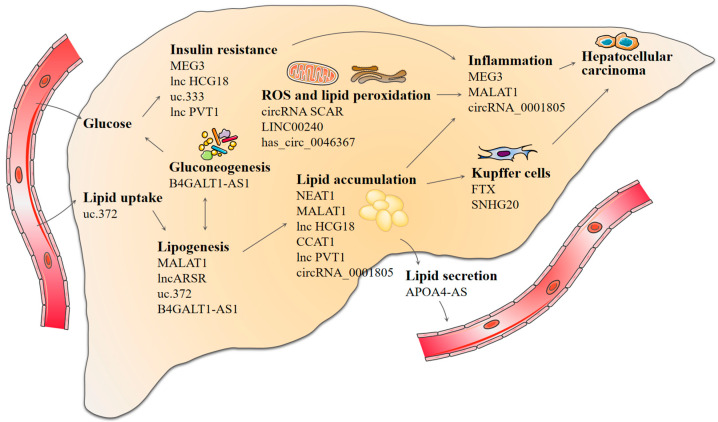
Pleiotropic roles of lncRNA and circRNA in the pathogenesis of NAFLD.

**Table 1 biomolecules-13-00560-t001:** LncRNA and circRNA Expression Profile in Patients with NAFLD/NAFL/NASH. “⤤”, “⤦”, “⟷” indicates the expression direction; “√”, means studies that used quantitative real-time PCR to measure lncRNA and/or circRNA expression.

NAFLD vs. Healthy Control									
lncRNA	Species	Expression Direction	Sample Size (NAFLD/Healthy Control)	Fold-Change	*p* Value	Possible Mechanism	Sequencing	qRT-PCR	Study
NEAT1	peripheral blood mononuclear cells	⤤	119/100	2.4	<0.05	-	-	√	Zhou, 2022 [27]
NEAT1	serum	⤤	10/10	5	<0.001	lncNEAT1/miR-212-5p/GRIA3	-	√	Hu, 2022 [24]
MEG3	liver	⤤	5/7	1.6	<0.05	managing obesity-associated hepatic endothelial senescence and insulin resistance	RNA sequencing	√	Cheng, 2021 [28]
MALAT1	liver	⤤	20/10	1.73	<0.05	MALAT1/miR-206/ARNT	-	√	Xiang, 2022 [29]
ENST00000413567	liver	⤦	5/5	−4.22 (0.05)	<0.05	-	Microarray	√	Sun, 2015 [30]
NR_026836	liver	⟷	5/5		>0.05	-	Microarray	√	Sun, 2015 [30]
ENST00000548691	liver	⤤	5/5	4.74 (26)	<0.05	-	Microarray	√	Sun, 2015 [30]
ENST00000571619	liver	⟷	5/5		>0.05	-	Microarray	√	Sun, 2015 [30]
uc001gvp.1	liver	⤦	5/5	−3.53 (0.09)	<0.05	-	Microarray	√	Sun, 2015 [30]
TCONS_00025733	liver	⤦	5/5	−1.07 (0.48)	<0.05	-	Microarray	√	Sun, 2015 [30]
ENST00000491676	liver	⟷	5/5		>0.05	-	Microarray	√	Sun, 2015 [30]
APOA4-AS	liver	⤤	8/5	8.3	<0.05	HuR–APOA4-AS complex stabilizes APOA4 mRNA	RNA sequencing	√	Qin, 2016 [31]
uc.372	liver	⤤	11/11	2.06	<0.001	suppressing miR-195/miR4668 maturation	Microarray	√	Guo, 2018 [20]
uc.94	liver	⤦	5/5	0.42	<0.05	-	Microarray	√	Guo, 2018 [20]
uc.157	liver	⤤	5/5	1.32	<0.05	-	Microarray	√	Guo, 2018 [20]
lncRNA B4GALT1-AS1(the human homologous sequence of mouse lncSHGL)	liver	⤦	6/6	0.74	<0.05	lncSHGL/hnRNPA1/CaM/Akt	Microarray	√	Wang, 2018 [32]
lncARSR	liver	⤤	12/22	3.76	<0.01	regulating hepatic lipogenesis via Akt/SREBP-1c pathway	-	√	Zhang, 2018 [26]
	serum	⤤	12/22	5.75	<0.01	-	-	√	Zhang, 2018 [26]
RP11-279F6.1	liver	⤤	2/2	2.27	<0.01	-	Microarray	√	Wu, 2019 [33]
AC004540.4	liver	⤤	2/2	2	<0.01	-	Microarray	√	Wu, 2019 [33]
uc.333	liver	⤦	8/8	0.13	<0.01	improving IR by binding to miR-223(FOXO1/AKT/GSK)	Microarray and lncRNA probe mapping	√	Zhang, 2020 [21]
CCAT1	liver	⤤	17/10	2.58	<0.05	CCAT1/miR-613/LXRα	Microarray	√	Huang, 2020 [22]
lncRNA RABGAP1LDT-206	serum	⤤	100/100		<0.001	-	Microarray	√	Albadawy, 2021 [34]
lnc-SPARCL1-1:2	serum	⤤	25/80	1.8	<0.05	lnc-SPARCL1-1:2/miR-6881-5P/HSPD1/MMP14/ITGB1	-	√	Albadawy, 2021 [23]
lnc PVT1	serum	⤤	81/78	1.46	<0.001	lncPVT1/miR-20a-5p	-	√	Zhang, 2021 [25]
lnc HCG18	serum	⤤	116/101	1.89	<0.001	HCG18-miR-197-3p	-	√	Xia, 2021 [35]
LINC00240	liver	⤤	3/3	5.25	<0.01	increasing the ROS content	GSE107231 dataset	√	Xue, 2022 [36]
has_circ_0046367	liver	⤦	5/3	0.4	<0.01	circRNA_0046367/miR-34a/PPARα	-	√	Guo, 2017 [37]
circRNA_0001805	liver	⤦	25/9	0.63	<0.05	circRNA_0001805 -miR-106a-5p/miR-320a-ABCA1/CPT1	High-throughput sequencing	√	Li, 2021 [38]
**NAFL vs. Healthy Control**									
**lncRNA**	**Species**	**Expression Direction**	**Sample Size (NAFL/Healthy Control)**	**Fold-Change**	***p* Value**	**Possible Mechanism**	**Sequencing**	**√**	**Study**
MEG3	liver	⟷	8/6	-	>0.05	-	-	√	Zhang, 2017 [39]
MEG3	liver	⤦	15/10	0.6	<0.05	MEG3 up-regulates SIRT6 by ubiquitinating EZH2	-	√	Zou, 2022 [40]
MALAT1	liver	⟷	32/13	-	0.9	-	Systems biology multiscale modeling	√	Sookoian, 2018 [41]
LncPRYP4-3	serum	⤤	30/30	10	<0.0001	targeting RPS4Y2	Microarray and lncRNA probe mapping	√	Yang, 2020 [42]
LncPRYP4-4	serum	⤤	30/30	9.6	<0.05	-	Microarray and lncRNA probe mapping	√	Yang, 2020 [42]
**NASH vs. Healthy Control**									
**lncRNA**	**Species**	**Expression Direction**	**Sample Size (NASH/Healthy control)**	**Fold-Change**	***p* Value**	**Possible Mechanism**	**Sequencing**	**√**	**Study**
MEG3	liver	⤤	7/7	1.9	<0.05	managing obesity-associated hepatic endothelial senescence and insulin resistance	RNA sequencing	√	Cheng, 2021 [28]
RP11-128N14.5	serum	⤤	45/25	2.33	≤0.05	-	Microarray and lncRNA probe mapping	√	Di Mauro, 2019 [43]
LncPRYP4-3	serum	⤤	30/30	4.7	<0.05	targeting RPS4Y2	Microarray and lncRNA probe mapping	√	Yang, 2020 [42]
LncPRYP4-4	serum	⟷	30/30	-	>0.05	-	Microarray and lncRNA probe mapping	√	Yang, 2020 [42]
lnc-SPARCL1-1:2	serum	⤦	55/80	-	<0.01	lnc-SPARCL1-1:2/miR-6881-5P/HSPD1/MMP14/ITGB1	-	√	Albadawy, 2021 [23]
lncRNA RABGAP1LDT-206	serum	⤤	60/100	-	<0.001	-	Microarray	√	Albadawy, 2021 [34]
circRNA SCAR	liver mitochondrial	⤦	18/20	0.16	<0.001	binding to ATP5B to inhibit mitochondrial ROS	Microarray	√	Zhao, 2020 [44]
hsa_circ_0089763	liver mitochondrial	⤦	18/20	0.28	<0.001	-	Microarray	√	Zhao, 2020 [44]
hsa_circ_0008882	liver mitochondrial	⤦	18/20	0.39	<0.001	-	Microarray	√	Zhao, 2020 [44]
hsa_circ_0073378	liver mitochondrial	⤦	18/20	0.68	<0.05	-	Microarray	√	Zhao, 2020 [44]
hsa_circ_0022430	liver mitochondrial	⤦	18/20	0.65	<0.01	-	Microarray	√	Zhao, 2020 [44]
hsa_circ_0032777	liver mitochondrial	⤦	18/20	0.53	<0.01	-	Microarray	√	Zhao, 2020 [44]
hsa_circ_0141729	liver mitochondrial	⤦	18/20	0.67	<0.05	-	Microarray	√	Zhao, 2020 [44]
hsa_circ_0001311	liver mitochondrial	⤦	18/20	0.74	<0.01	-	Microarray	√	Zhao, 2020 [44]
hsa_circ_0045989	liver mitochondrial	⤦	18/20	0.56	<0.05	-	Microarray	√	Zhao, 2020 [44]
**NASH vs. NAFL**									
**lncRNA**	**Species**	**Expression Direction**	**Sample Size (NASH/NAFL)**	**Fold-Change**	***p* Value**	**Possible Mechanism**	**Sequencing**	**√**	**Study**
MEG3	liver	⤦	6/9	0.48	<0.05	MEG3 up-regulates SIRT6 by ubiquitinating EZH2	-	√	Zou, 2022 [40]
MALAT1	liver	⤤	15/32	1.75	0.029	-	Systems biology multiscale modeling	√	Sookoian, 2018 [41]
LncPRYP4-3	serum	⤦	30/30	0.47	<0.05	targeting RPS4Y2	Microarray and lncRNA probe mapping	√	Yang, 2020 [42]
LncPRYP4-4	serum	⟷	30/30	-	>0.05	-	Microarray and lncRNA probe mapping	√	Yang, 2020 [42]
LeXis	plasma	⤤	35/9	1.75	0.025	-	-	√	Park, 2020 [45]
	liver	⟷	35/9		0.539	-	-	√	Park, 2020 [45]
lnc-SPARCL1-1:2	serum	⤤	55/11	1.13	<0.01	lnc-SPARCL1-1:2/miR-6881-5P/HSPD1/MMP14/ITGB1	-	√	Albadawy, 2021 [23]

**Table 2 biomolecules-13-00560-t002:** LncRNA Expression is Associated with NASH Phenotype. “⤤”, “⤦”, “⟷” indicates the expression direction.

Steatosis						
lncRNA	Species	Sample Size	Expression	Fold-Change	*p* Value	Study
GAS5	liver	30/21	⟷	-	0.602	Han, 2020 [49]
	plasma	30/21	⟷	-	0.274	Han, 2020 [49]
LeXis	liver	35/9	⤦	0.49	0.017	Park, 2020 [45]
LeXis	plasma	35/9	⟷	-	0.399	Park, 2020 [45]
**Lobular Inflammation**						
**lncRNA**	**Species**	**Sample Size (with/without)**	**Expression**	**Fold-Change**	***p* Value**	**Study**
MALAT1	liver	34/15	⤤	1.91	0.0025	Sookoian, 2018 [41]
MALAT1	liver	53/24	⟷	-	>0.05	Leti, 2017 [47]
NEAT1	liver	53/24	⤤	1.33	1.0 × 10^−4^	Leti, 2017 [47]
HULC	liver	53/24	⤤	2.3	2.53 × 10^−5^	Leti, 2017 [47]
lncTNF	liver	35/9	⤤	R = 0.58	9.7 × 10^−7^	Atanasovska, 2021 [48]
LeXis	liver	35/9	⟷	-	0.914	Park, 2020 [45]
LeXis	plasma	35/9	⟷	-	0.404	Park, 2020 [45]
lnc18q22.2	liver	17/8	⤤	R = 0.62	1.38 × 10^−4^	Atanasovska, 2017 [46]
GAS5	liver	29/32	⟷	-	0.602	Han, 2020 [49]
	plasma	29/32	⟷	-	0.274	Han, 2020 [49]
**Ballooning Degeneration**						
**lncRNA**	**Species**	**Sample Size (with/without)**	**Expression**	**Fold-Change**	***p* Value**	**Study**
MALAT1	liver	34/15	⤤	3.01	0.0001	Sookoian, 2018 [41]
**NAS Score**						
**lncRNA**	**Species**	**Sample Size**	**Expression**	**Fold-Change**	***p* Value**	**Study**
**NAS > 5 vs. HC**						
RP11-128N14.5	serum	25/25	⤤	2.69	<0.02	Di Mauro, 2019 [43]
TGFB2/TGFB2-OT1	serum	25/25	⤤	26.1	<0.05	Di Mauro, 2019 [43]
**NAS>5 vs. NAS<=4**						
RP11-128N14.5(internal)	serum	25/38	⤤	1.25	<0.05	Di Mauro, 2019 [43]
TGFB2/TGFB2-OT1(internal)	serum	25/38	⤤	1.58	<0.05	Di Mauro, 2019 [43]
RP11-128N14.5(external)	serum	27/23	⤤	4.3	0.04	Di Mauro, 2019 [43]
TGFB2/TGFB2-OT1(external)	serum	27/23	⤤	6.2	0.03	Di Mauro, 2019 [43]
LeXis	liver	35/9	⟷	-	0.872	Park, 2020 [45]
	plasma	35/9	⟷	-	0.363	Park, 2020 [45]
lnc18q22.2	liver	17/8	⤤	R = 0.58	8.64 × 10^−4^	Atanasovska, 2017 [46]
GAS5	liver	24/27	⟷	-	0.674	Han, 2020 [49]
	plasma	24/27	⟷	-	0.448	Han, 2020 [49]
**NASH Grade**						
**lncRNA**	**Species**	**Sample Size (with/without)**	**Expression**	**Fold-Change**	***p* Value**	**Study**
lnc18q22.2	liver	17/8	⤤	0.65	4.55 × 10^−4^	Atanasovska, 2017 [46]

**Table 3 biomolecules-13-00560-t003:** LncRNA Expression Profile in Patients with NAFLD with Different Fibrosis Stages. “⤤”, “⤦”, “⟷” indicates the expression direction.

Fibrosis	lncRNA	Species	Sample Size	Expression	Fold-Change	*p* Value	Study
F0 vs. F1–4	MALAT1	liver	13 HC/47 NAFLD	⤤	5	1 × 10^−7^	Sookoian, 2018 [41]
	MEG3	liver	6 HC/6 fibrosis	⤤	2.2	<0.01	Zhang, 2017 [39]
	MEG3	liver	10HC/15 fibrosis	⤦	0.3	<0.01	Zou, 2022 [40]
F0–2 vs. F3	GAS5	liver	39 F0-2/6 F3	⟷	-	0.131	Han, 2020 [49]
	GAS5	plasma	39 F0-2/6 F3	⤤	2.02	<0.001	Han, 2020 [49]
F0–2 vs. F3–4	TGFB2/TGFB2-OT1	serum	37 F0-2/26 F3-4	⤤	1.82	≤0.001	Di Mauro, 2019 [43]
	RP11-128N14.5	serum	37 F0-2/26 F3-4	⟷	-	>0.05	Di Mauro, 2019 [43]
F0–1 vs. F4	lnc-SPARCL1-1:2	serum	25 F0-1/10 F4	⤤	-	<0.05	Albadawy, 2021 [23]
	lncRNA RABGAP1LDT-206	serum	34 F3/11 F4	⤤	-	<0.05	Albadawy, 2021 [34]
F2 vs. F4	lnc-SPARCL1-1:2	serum	20 F2/10 F4	⤤	-	<0.05	Albadawy, 2021 [23]
	lncRNA RABGAP1LDT-206	serum	26 F2/11 F4	⤤	-	<0.05	Albadawy, 2021 [34]
F3 vs. F4	GAS5	liver	6 F3/6 F4	⟷	-	0.818	Han, 2020 [49]
	GAS5	plasma	6 F3/6 F4	⤦	0.54	0.026	Han, 2020 [49]
	lnc-SPARCL1-1:2	serum	24 F3/10 F4	⤤	-	<0.05	Albadawy, 2021 [23]
	lncRNA RABGAP1LDT-206	serum	29 F3/11 F4	⤤	-	<0.05	Albadawy, 2021 [34]
(advanced fibrosis) vs. (not advanced fibrosis)	LeXis	serum	33/11	⟷	-	0.328	Park, 2020 [45]
		liver	33/11	⟷	-	0.14	Park, 2020 [45]
	NEAT1	liver	24/53	⤤	1.29	3 × 10^−4^	Leti, 2017 [47]
	HULC	liver	24/53	⤤	3.6	1.08 × 10^−8^	Leti, 2017 [47]
	MALAT1	liver	24/53	⤤	3.6	8.02 × 10^−6^	Leti, 2017 [47]
	LINC01638	liver	10/10	⤤	2.81	≤0.001	Gerhard, 2020 [51]
	LINC01605	liver	10/10	⟷	-	>0.05	Gerhard, 2020 [51]
	XLOC_003146	liver	10/10	⤤	1.65	≤0.0001	Gerhard, 2020 [51]
	RP11_20J153	liver	10/10	⤤	1.34	≤0.0001	Gerhard, 2020 [51]
HC vs. cirrhosis	GAS5	liver	15/20	⤦	0.43	<0.01	Yu, 2015 [50]
	MEG3	liver	6/8	⤤	2.2	<0.01	Zhang, 2017 [39]
	H19	liver	6/17	⤤	3.67	<0.05	Liu, 2018 [52]

## Data Availability

Not applicable.

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
