# Peer review of "LncRNA and circRNA in Patients with Non-Alcoholic Fatty Liver Disease: A Systematic Review"

_biomolecules, 2023, doi:10.3390/biom13030560_

Round 1

Reviewer 1 Report

Blind Comments to the Authors

The manuscript ” LncRNA and circRNA in patients with non-alcoholic fatty liver disease: A systematic review “ by Qingmin Zeng et al. summarizes and discusses the role of long non-coding RNAs (LncRNA) and circular RNAs (circRNA) in steatotic related liver diseases. The focus of the review is a systematically analysis of the existing literature to verify the clinical value of both non coding RNAs. In general, non-coding RNAs are an important topic in a field of increasing and broad interest. The strength of the presented manuscript is its clear methodological way of reviewing the existing literature and the structured way of the presentation of its results. Identified weakness and major and minor concerns in detail in the following:

Major concerns

Line 55-58 and 65-67: The functional role of ncRNAs is still unclear for non-experts. Therefore the introduction would benefit clearly if the most important functional and mechanistic ones were presented and related literature is cited to show that ncRNAs are no artificial macromolecules.

Figure 2 is designed confusing. The disease progressions follows NAFL, NAFLD, NASH, Fibrosis/cirrhosis. Looks like NAFLD was extracted here and shown in higher detail. But this is not easy to understand. While the upper boxes have headers the lower has none. The arrow in the middle could be for all shown boxes or only as description for the lower one. Please make this more clear and easy to understand. Additionally, the reference to Figure 2 in line 111 makes no sense because the figure content is about ncRNA markers and the text with the reference is about the studies. Please correct the placing of the figure reference.  

Page 14 1st paragraph: In contrast, Zou et al. showed […] provide more evidence. Beside the first sentence which has a citation, this paragraph looks like it misses references. For example “MEG3 has been found to maintain glucose homeostasis and insulin signaling by protecting the hepatic endothelium against cellular senescence in obesity.” Sounds like a citation. It is difficult to understand here what is citation, interpretation of literature and conclusion. Please revise.

Page 21: “Taken together, MALAT1 and GAS5 were positively correlated with NASH grade and/or fibrosis, which may be candidate biomarkers for severe NAFLD phenotype and may open a new frontier for NAFLD drug development.”
There is a clear separation and are definitions of NAFLD and NASH. Here NASH and fibrosis are described as part of NAFLD which is not correct. Please revise. Please check the whole manuscript for the correct usage of the terms NAFL, NAFLD, NASH

Minor concerns

Line 47-48: To my knowledge severe fibrosis and cirrhosis are major indicators for liver transplantation (LTX). Please change or cite literature with valid numbers for NASH related LTX

Line 52-54: Statins and fibrates were considered for treatment of NAFLD. Why these options were not named here? Please discuss.

Line 58-59: Please add 1-3 good Reviews for the interested reader.

Line 75: The abbreviation NAFL was not introduced and defined. NAFL as the benign form of NAFLD should be listed first to reflect the progression pathway.

Line 123, page 20-22. There are 4 java script references shown by text in a different style. Please remove.

Page 15: “However, the number of patients included in cu rrent studies was rela-tively small.” Please correct current

Taken together this review is an interesting read and a valuable summery of knowledge about some NAFLD related ncRNAs. The presented manuscript shows some weakness in consistent terminology and needs some corrections and clarifications. Therefore, I recommend accepting after minor revision.

Reviewer 2 Report

In this review manuscript by Zeng et al, the authors systematically summarized the recent advances in studies in lncRNAs and circRNAs and their involvement in the development of NAFLD. A number of lncRNAs with diagnostic potential were also selected for discussion. This is a well-constructed review. However, there are a number of concerns that need to be addressed before this manuscript is in a publishable fashion. Specific comments are as follows:  

1) The main topic of this review is about lncRNAs and circRNAs. However, there are only two major articles about circRNAs. It is suggested to leave out the part for circRNAs or rearrange into a separate section so that its importance can stand out.  

2) As in 1), the corresponding titles in subsections also need to be changed as some of the sections do not contain circRNA studies.  

3) The rationale of selecting some of the lncRNAs as these may have diagnostic value needs to be strengthened. For example, the results from different groups in MEG3 studies are inconsistent. Both MEG and MALAT1 have only tested in the liver tissue so far but not in the circulation. These may further need to be justified.  

4) Some of the statements are too strong and do not seem to have sufficient references behind them. For example, in the fibrosis/cirrhosis section, "those lncRNAs enable us to discriminate NAFLD/NASH cases from advanced fibrosis". There is only one lncRNA (MALAT1) described in this article and the conclusion (in the discussion) was based on two separate studies. Another example is in the HCC section, "most HCC-related lncRNAs are present in body fluids, which are easy to detect and analyze". This statement also needs references as the two articles reviewed in this section do not suggest this.
